# Constrained Optimization From a Control Perspective via Feedback Linearization

## Abstract

Constrained optimization is fundamental to numerous applications. While first-order iterative algorithms are widely used for solving these problems, understanding their continuous-time counterparts—formulated as differential equations—can provide valuable theoretical insights into stability and convergence. Among various approaches, Feedback Linearization (FL), a well-established nonlinear control technique, has demonstrated potential for addressing nonconvex equality-constrained optimization problems, yet remains relatively underexplored.

This paper aims to develop rigorous theoretical foundations for applying feedback linearization to solve constrained optimization. For equality-constrained optimization, we establish global convergence rates to first-order Karush-Kuhn-Tucker (KKT) points and uncover the close connection between the FL method and the Sequential Quadratic Programming (SQP) algorithm. Building on this relationship, we extend the FL approach to handle inequality-constrained problems. Furthermore, we introduce a momentum-accelerated FL algorithm that achieves faster convergence, and provide a rigorous convergence guarantee.

## 1. Introduction

Constrained optimization, also known as nonlinear programming, has found vast applications in several domains including robotics (Alonso-Mora et al., 2017), supply chains (Garcia and You, 2015), and safe operations of power systems (Dommel and Tinney, 1968). First-order iterative algorithms are widely used to solve such problems, particularly in optimization and machine learning settings with large-scale datasets. These algorithms can be interpreted as discrete-time (DT) dynamical systems, while their continuous-time (CT) counterparts, derived by con-

sidering infinitesimal step sizes, take the form of differential equations. Analyzing these continuous-time systems can provide valuable theoretical insights, such as stability properties and convergence rates. This perspective is well-developed for unconstrained optimization, exemplified by the gradient flow $\dot{x} = -\nabla f(x)$ (Elkabetz and Cohen, 2021; Arora et al., 2019; Saxe et al., 2013; Garg and Panagou, 2021; Andrei), the continuous-time counterpart of gradient descent, as well as its accelerated variants (Su et al., 2016; Wilson et al., 2018; Muehlebach and Jordan, 2019). However, for constrained optimization, this approach remains less thoroughly explored.

Recent studies (Cerone et al., 2024; Gunjal et al., 2024) have explored the dynamical properties of CT constrained optimization algorithms. These works leverage a feedback control perspective to design and analyze the performance of optimization methods. Specifically, they propose frameworks that model constrained optimization problems as control problems, where the iterations of the optimization algorithm are represented by a dynamical system, and the Lagrange multipliers act as control inputs. The objective in this framework is to drive the system to a feasible steady state that satisfies the constraints. Within this framework, various control strategies can be employed to design the update of Lagrange multipliers, resulting in different control-based first-order methods.

In this work, we adopt the same control perspective as above, and specifically focus on using Feedback Linearization (FL) approach, a standard approach in nonlinear control (cf. (Isidori, 1985; Henson and Seborg, 1997)), to design the Lagrange multiplier. One key advantage of this method is its natural suitability for handling nonconvex constrained optimization problems. Although this approach has been explored in the literature (Cerone et al., 2024; Schropp and Singer, 2000), its theoretical properties are not yet fully understood. Several important questions remain open.

The first question concerns global convergence and convergence rates. While existing works established local stability (Cerone et al., 2024), global convergence and convergence rate have not been rigorously analyzed. The second question concerns the relationship between the FL ap-

Preliminary work. Under review by the International Conference on Machine Learning (ICML). Do not distribute.

proach and existing optimization algorithms, specifically whether the discretization of FL dynamics aligns with any known optimization method. Additionally, since most existing studies (Cerone et al., 2024; Schropp and Singer, 2000) focus exclusively on equality constraints, this raises the third question: how can the FL approach be extended to effectively handle inequality constraints? Lastly, it remains an open question whether insights from acceleration techniques in optimization (e.g., momentum acceleration) can be leveraged to develop faster FL-based algorithms.

**Our contributions.** Motivated by the open questions discussed above, we aim to deepen the theoretical understanding of the feedback linearization (FL) approach for constrained optimization by addressing these questions. Specifically, our contributions are as follows:

1. We establish a global convergence rate to a first-order Karush-Kuhn-Tucker (KKT) point for the feedback linearization method for equality-constrained optimization (Section 3.1).

2. We demonstrate that the FL method is closely related to the Sequential Quadratic Programming (SQP) algorithm, providing a new perspective on its connection to established optimization techniques (Section 3.2).

3. Building on this insight, we extend the method to handle inequality constraints, broadening its applicability (Section 4).

4. Finally, leveraging these findings, we propose a momentum-accelerated FL algorithm, which empirically achieves accelerated convergence. Additionally, we provide a rigorous convergence guarantee for the continuous-time momentum-accelerated FL method (Section 5). To the best of our knowledge, both the proposed algorithm and its analysis are novel contributions to the field.

Due to space limits, a comprehensive review of related literature is deferred to Appendix A.

**Notations:** We use the notation $[n], n \in \mathbb{N}$ to denote the set $\{1, 2, 3, \ldots, n\}$. We use $\nabla f(x)$ to denote the gradient of a scalar function $f : \mathbb{R}^n \to \mathbb{R}$ evaluated at the point $x \in \mathbb{R}^n$ and use $\nabla^2 f(x)$ to denote its corresponding Hessian matrix. We use $J_h(x)$ to denote the Jacobian matrix of a function $h : \mathbb{R}^n \to \mathbb{R}^m$ evaluated at $x \in \mathbb{R}^n$, i.e. $[J_h(x)]_{i,j} = \frac{\partial h_i(x)}{\partial x_j}$, $i \in [m], j \in [n]$. Unless specified otherwise, we use $\| \cdot \|$ to denote the $L_2$ norm of matrices and vectors and use $\| \cdot \|_\infty$ to denote the $L_\infty$ norm. For a positive definite matrix $A$, we use $\|X\|_A := \|A^{-\frac{1}{2}} X\|$ to denote the $A$-norm of $X$. For a set $\mathcal{A}$, we use $\mathcal{A}^c$ to denote its complement.

## 2. Feedback Linearization for solving equality constrained optimization

In this section, we briefly review related works that adopt a control perspective, particularly focusing on the use of feedback linearization to address equality-constrained optimization problems.

**Control perspective on equality-constrained optimization (Cerone et al., 2024)** Consider the constrained optimization problem with equality constraints

$$\min_x f(x) \qquad s.t. \ h(x) = 0, \qquad (1)$$

where $x \in \mathbb{R}^n$, $f : \mathbb{R}^n \to \mathbb{R}, h : \mathbb{R}^n \to \mathbb{R}^m$. The first-order KKT conditions are given by

$$-\nabla f(x) - J_h(x)^\top \lambda = 0, \quad h(x) = 0 \qquad (2)$$

The key idea is to view finding the KKT point as a control problem (Figure 1)

$$\dot{x} = -T(x) \left( \nabla f(x) + J_h(x)^\top \lambda \right), \quad y = h(x) \qquad (3)$$

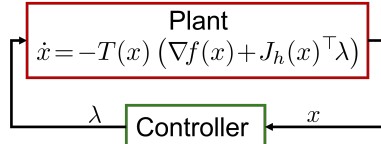

Figure 1: Control Perspective for Constrained Optimization

where $x$ represents the system state, $y = h(x)$ is system constraint variable and $\lambda$ is the control input. $T(x)$ here is a positive definite matrix and throughout the paper we assume that there exists $\lambda_{\min}, \lambda_{\max}$ such that for all $x$,

$$\lambda_{\min} I \preceq H(x) \preceq \lambda_{\max} I$$

Note that at an equilibrium point $x^\star$ of the system in Fig. 1 it must satisfy:

$$\dot{x} = 0 \implies \nabla f(x^\star) + J_h^\top(x^\star) = 0.$$

Further, if $x^\star$ is feasible, i.e. $h(x^\star) = 0$, then we get that $x^\star$ satisfies the first order KKT conditions (2). Thus, the key idea is to manipulate the evolution of $x$ so that we stabilize the system to equilibrium and feasibility. (For a more detailed overview about optimization from a control perspective, see Appendix A.)

To achieve the goal of reaching a feasible equilibrium, we next introduce the feedback linearization (FL) approach, which is the main focus of this paper.

**Feedback linearization for equality-constrained optimization (Cerone et al., 2024)** Feedback linearization (FL) (Isidori, 1985; Henson and Seborg, 1997) is a classical control method for controlling nonlinear dynamics which generally takes the following form:

$$\dot{x} = F(x) + G(x)\lambda$$

As directly designing a stabilizing controller for the nonlinear system can be a challenging task, the FL approach circumvents the difficulty by transforming a nonlinear control system into an equivalent linear control system, which is much easier to analyze, through a change of variables and a suitable control input. In particular, one seeks a change of coordinates $y = \Phi(x)$ and the control input $\lambda = a(x) + b(x)u$ such that the system becomes a linear system: $\dot{y} = Ay + Bu$.

In the equality constrained optimization problem, we have that $F(x) = -\nabla f(x), G(x) = -J_h(x)$. The difference is that we don't seek a bijective change of coordinates $y = \Phi(x)$, but only focus on the observations $y = h(x)$ instead. If $\lambda = a(x) + b(x)u$ and we write out the time derivative for $y$ we have that

$$\dot{y} = J_h(x)\dot{x} = -J_h(x)T(x)\nabla f(x) - J_h(x)T(x)J_h(x)^\top\lambda$$
$$= -J_h(x)T(x)\nabla f(x) - J_h(x)T(x)J_h(x)^\top(a(x) + b(x)u)$$

Thus, in the scenario where $J_h(x)$ has full row rank, by setting

$$a(x) = -\left(J_h(x)T(x)J_h(x)^\top\right)^{-1} J_h(x)T(x)\nabla f(x),$$
$$b(x) = -\left(J_h(x)T(x)J_h(x)^\top\right)^{-1},$$

we have that $\dot{y} = u$, then we can simply set $u = -Ky$ where $K$ is a Hurwitz matrix to guarantee that $y$ asymptotically converge to zero. Thus the feedback linearization (FL) dynamics is given as follows:

---
**FL for Equality-Constrained Optimization (Cerone et al., 2024)**

$$\dot{x} = -T(x)\left(\nabla f(x) + J_h(x)^\top\lambda\right) \quad (4)$$
$$\lambda = -\left(J_h(x)T(x)J_h(x)^\top\right)^{-1}(J_h(x)T(x)\nabla f(x) - Kh(x))$$

---

The FL approach is particularly advantageous for handling nonlinear dynamics, making it well-suited for nonconvex constrained optimization. This is supported by numerical results in (Cerone et al., 2024; Schropp and Singer, 2000), which demonstrate its strong performance in such settings. However, its theoretical properties remain less well understood. Existing analyses primarily focus on local stability (Cerone et al., 2024), while global convergence and convergence rates are largely unexplored. Furthermore, the relationship between the FL algorithm and existing optimization methods remains unclear.

The following Section 3 will focus on addressing the above open problems in the FL method that remain unsolved in the existing literature, including convergence rate (Section 3.1) and relationship to existing optimization algorithms (Section 3.2)

## 3. FL control method: convergence and Relationship to SQP

### 3.1. Convergence Results

Section 2 introduces the FL method for equality-constrained optimization. The analyses in existing works mainly focus on the local stability, and little is known in terms of the global convergence property. In this section, we establish a global convergence rate to a first order KKT point (Contribution 1).

The result relies on the following assumption:

**Assumption 1.** *We make the following assumptions on the function $f$ and $h$:*

*1.1 There exists a constant $D$ such that $(J_h(x)J_h(x)^\top)^{-1} \prec D^2 I$ for all $x$;*

*1.2 There exists a constant $M$ such that $\|\nabla f(x)\| < M$, $\|J_h(x)\| < M$ for all $x$;*

*1.3 The function $f(x)$ is lower-bounded, i.e. $f(x) \geq f_{\min}$ for all $x$.*

Note that Assumption 1.1 is similar to the assumption made in (Cerone et al., 2024) which assumes that $\text{rank}(J_h(x)) = m$ for all $x$, which is equivalent to $J_h(x)J_h(x)^\top$ is invertible. This assumption is also known as the linear independence constraint qualification (LICQ, cf. (Peterson, 1973; Nocedal and Wright, 2006), see more discussion in Appendix A) in optimization literature. Assumption 1.2 implies that the functions $f$ and $g$ are Lipschitz. We would also like to acknowledge that this Assumption is quite restrictive and is solely for analysis purpose. In our numerical simulations we found that the algorithm is suitable for non-uniformly-Lipschitz functions.

We define the KKT-gap of $(x, \lambda)$ as follows:

$$\texttt{KKT-gap}(x,\lambda) := \max\{\|\nabla f(x) + J_h(x)^\top\lambda\|, \|h(x)\|_\infty\} \quad (5)$$

We now state our result in terms of the convergence rate

**Theorem 1.** *Under Assumption 1, for control gain $K$ that is a diagonal positive definite matrix, i.e., $K =$*

---
We note that if $x^\star \in D$ is known a priori for a compact domain $D$, a potential approach for handling non-uniformly Lipschitz functions $f, g$ is to construct smooth and Lipschitz extensions $f', g'$ such that their gradients and Jacobians match those of $f, g$ within $D$ while remaining uniformly Lipschitz outside $D$ (cf. (Stein, 1970)).

diag$\{k_i\}_{i=1}^m$, *where $k_i > 0$, we have that the dynamic of feedback linearization* (4) *satisfies:*

1. *For the set $\mathcal{E}_i := \{x : h_i(x) \geq 0\}$, if $x(0) \in \mathcal{E}_i$, then $x(t) \in \mathcal{E}_i$ for all $t \geq 0$. Similarly, if $x(0) \in \mathcal{E}_i^c$, then $x(t) \in \mathcal{E}_i^c$ for all $t \geq 0$, further*

$$h_i(x(t)) = e^{-k_i t} h_i(x(0)),$$

   *i.e., $h(x(t)) \to 0$ with an exponential rate as $t \to +\infty$.*

2. *Define $\ell(x) := f(x) + \frac{\lambda_{\max}}{\lambda_{\min}}(MD)^2 \sum_{i=1}^m |h_i(x)|$, then $\ell(x(t))$ is non-increasing with respect to $t$.*

3. *Let $\overline{\lambda}(t) := -\left(J_h(x)T(x)J_h(x)^\top\right)^{-1} J_h(x)T(x)\nabla f(x(t))$, then we have that*

$$\int_{t=0}^{T} \|\nabla f(x(t)) + J_h(x(t))^\top \overline{\lambda}(t)\|^2 dt \leq \frac{1}{\lambda_{\min}}\left(\ell(x(0)) - \ell(x(T))\right),$$

   *and that $\lim_{t\to+\infty}\left(\lambda(t) - \overline{\lambda}(t)\right) = 0$.*

4. *(Asymptotic convergence and convergence rate) The above statements imply that,*

$$\inf_{0 \leq t \leq T} \texttt{KKT-gap}(x(t), \overline{\lambda}(t)) \leq$$

$$\max\left\{\sqrt{\frac{2}{T}\left(\frac{f(x(0)) - f_{\min}}{\lambda_{\min}} + \frac{\lambda_{\max}M^2D^2}{\lambda_{\min}^2}\sum_i |h_i(x(0))|\right)},\right.$$

$$\left.\max_{1 \leq i \leq m}\left\{h_i(x(0))e^{-\frac{k_i T}{2}}\right\}\right\} \sim O\left(\frac{1}{\sqrt{T}}\right)$$

   *further, we have that*

$$\lim_{t\to+\infty} \texttt{KKT-gap}(x(t), \overline{\lambda}(t)) = 0,$$

$$\lim_{t\to+\infty} \texttt{KKT-gap}(x(t), \lambda(t)) = 0.$$

Statement 4 in Theorem 1 implies that the algorithm can find an $\epsilon$-first-order-KKT-point within time $\frac{1}{\epsilon^2}$. We note that ensuring last-iterate convergence in nonconvex optimization is generally challenging. Hence, our analysis focuses on the best iterate, a widely adopted criterion in nonconvex optimization. Due to space limitations, we defer the detailed proof in Appendix B. The key step of the proof involves constructing the Lyapunov function $\ell(x)$ in Statement 2. We would also like to note that $\ell(x)$ also serves as the exact penalty function in constrained optimization literature (cf. (Eremin, 1967; Zangwill, 1967)).

### 3.2. Relationship with SQP

The FL dynamics (4) provides a concise and elegant formulation, prompting the question of whether certain optimization algorithms can be derived through its discretization. In this section, we establish a fundamental connection between the continuous-time FL dynamics and the Sequential Quadratic Programming (SQP) algorithm (Contribution 2). Specifically, we demonstrate that the forward-Euler discretization (cf. (Atkinson, 1991; Ascher and Petzold, 1998)) of (4) is equivalent to the SQP algorithm.

The state space continuous time dynamic for (4) is

$$\dot{x} = -T(x)\Big(\nabla f(x) - J_h(x)^\top \left(J_h(x)T(x)J_h(x)^\top\right)^{-1} \\ \cdot \left(J_h(x)T(x)\nabla f(x) - Kh(x)\right)\Big).$$

Its forward-Euler discretization scheme is

$$x_{t+1} = x_t - \eta T(x_t)\Big(\nabla f(x_t) - J_h(x_t)^\top \quad (6) \\ \cdot \left(J_h(x_t)T(x_t)J_h(x_t)^\top\right)^{-1}(J_h(x_t)T(x_t)\nabla f(x_t) - Kh(x_t))\Big).$$

We now consider the following SQP method, which is widely discussed in literature (cf. (Nocedal and Wright, 2006; Bonnans et al., 2006; Oztoprak et al., 2021)):

$$x_{t+1} = \arg\min_x \nabla f(x_t)^\top(x - x_t) + \frac{1}{2\eta}(x - x_t)^\top T(x_t)^{-1}(x - x_t)$$

$$s.t. \quad h(x_t) + J_h(x_t)(x - x_t) = 0 \quad (7)$$

We are now ready to state the main result of this section, which demonstrates the equivalence of (6) and (7)

**Theorem 2.** *When $K = \frac{1}{\eta}I$, the discretization of feedback linearization (6) is equivalent to the SQP algorithm (7).*

The proof of Theorem 2 leverages the fact that (7) satisfies the relaxed Slater condition, and thus the KKT conditions are necessary and sufficient for global optimality. Then Theorem 2 can be obtained by studying the KKT conditions of 7. The detailed proof is deferred to Appendix C

**Remark 1** (Choice of $T(x)$)**.** *Theorem 2 provides insights into the selection of $T(x)$ for the FL approach. Different choices of $T(x)$ will correspond to different types of SQP algorithms. Here we mainly discuss two specific types of $T(x)$. Firstly, when $T(x)$ is chosen as the inverse of the Hessian matrix, i.e., $T(x) = \left(\nabla^2 f(x)\right)^{-1}$, then (7) corresponds to the Newton-type algorithm where the quadratic term in the objective function is given by $(x - x_t)^\top \nabla^2 f(x)(x - x_t)$, which is widely considered in literature (cf. (Nocedal and Wright, 2006; Bonnans et al., 2006)). For this specific type of $T(x)$, we name its corresponding FL dynamics (4) as the* **FL-Newton** *method. However, in the setting where the Hessian information is not available, another choice of $T(x)$ is simply setting it as the identity matrix $T(x) = I$, which is considered in recent works such as (Oztoprak et al., 2021). In this case, the objective function resembles a proximal operator (cf. (Boyd, 2004; Parikh et al., 2014)), hence we name this as* **FL-proximal** *method. Due to space limit, we defer a more comprehensive overview of SQP in to Appendix A.*

## 4. Extension to inequality constraints

The above sections primarily focus on the constrained optimization setting with equality constraints (1). This section aims to address the question of whether we can extend to setting with inequality constraints (Contribution 3), i.e.,

$$\min_x f(x) \qquad s.t. \ \ h(x) \le 0, \tag{8}$$

The KKT conditions for the above problem are given by

$$-\nabla f(x) - J_h(x)^\top \lambda = 0, \quad h(x) \le 0$$
$$\lambda \ge 0, \quad \lambda^\top h(x) = 0 \tag{9}$$

Thus, we can still view the problem as a control problem whose corresponding dynamics can be written as:

$$\dot{x} = -T(x)\left(\nabla f(x) + J_h(x)^\top \lambda\right)$$
$$y = h(x), \quad \lambda \ge 0. \tag{10}$$

However, the problem becomes more complicated because we require the non-negativity constraints $\lambda \ge 0$ and complementary slackness $\lambda^\top h(x) = 0$. It is at first glance unclear how to guarantee theses conditions through the control process. However, inspired by the relationship with SQP algorithms, we carefully design a more intricate FL controller as follows:

---

**FL for Inequality-Constrained Optimization**

$$\dot{x} = -T(x)\left(\nabla f(x) + J_h(x)^\top \lambda\right)$$
$$\lambda = \arg\min_{\lambda \ge 0}\left(\frac{1}{2}\lambda^\top J_h(x)T(x)J_h(x)^\top \lambda \right. \tag{11}$$
$$\left. + \lambda^\top \left(J_h(x)T(x)\nabla f(x) - Kh(x)\right)\right)$$

---

Here we assume that the optimization problem

$$\lambda = \arg\min_{\lambda \ge 0}\left(\frac{1}{2}\lambda^\top J_h(x)T(x)J_h(x)^\top \lambda \right.$$
$$\left. + \lambda^\top \left(J_h(x)T(x)\nabla f(x) - Kh(x)\right)\right)$$

admits a unique solution. We would also like to point out that $\lambda$ in (11) takes the form of the solution of an optimization problem, resulting in a non-smooth trajectory. A similar formulation of non-smooth ordinary differential equations (ODEs) has been explored in the context of differential variational inequalities (cf. (Dupuis and Nagurney, 1993; Pang and Stewart, 2008; Camlibel et al., 2007)).

At first glance, it may not be immediately clear why the algorithm is structured as in (11). The derivation of (11) was inspired by the connection between the FL method and SQP in the equality-constrained setting. Hence, for the inequality-constrained case, we first analyzed SQP and then reverse-engineered its principles to derive its

continuous-time counterpart, leading to the formulation of the FL method in (11). To ensure a coherent and intuitive presentation, we begin by establishing its relationship with the SQP algorithm.

**Relationship with the SQP algorithm** The corresponding forward Euler discretization of (11) is given by

$$x_{t+1} = x_t - \eta T(x)\left(\nabla f(x_t) + J_h(x_t)^\top \lambda_t\right)$$
$$\lambda_t = \arg\min_{\lambda \ge 0}\left(\frac{1}{2}\lambda^\top J_h(x_t)T(x_t)J_h(x_t)^\top \lambda \right. \tag{12}$$
$$\left. + \lambda^\top \left(J_h(x_t)T(x_t)\nabla f(x_t) - Kh(x_t)\right)\right)$$

We now consider the following SQP type of optimization method

$$x_{t+1} = \arg\min_x \nabla f(x_t)^\top (x - x_t) + \frac{1}{2\eta}(x - x_t)^\top T(x_t)^{-1}(x - x_t)$$
$$s.t. \quad h(x_t) + J_h(x_t)(x - x_t) \le 0 \tag{13}$$

The following theorem states the equivalence between (12) and (13).

**Theorem 3.** *When $K = \frac{1}{\eta}I$, if* (13) *is feasible, then the discretization of feedback linearization* (12) *is equivalent to the SQP algorithm* (13)*.*

Similar to the proof of Theorem 2, the proof of Theorem 3 also leverages strong duality and KKT conditions. THe detailed proof is deferred to Appendix C.

**Convergence Result** Theorem 3 demonstrates the relationship between the FL algorithm (11) and (13). Since SQP algorithms are known to be capable of converging to a KKT point (Nocedal and Wright, 2006), intuitively similar convergence can be established for our FL algorithm (11), which is the main focus of the following part.

We define the index set $\mathcal{I}(x) := \{i : h_i(x) > 0\}$. We also use $\mathcal{I}(x)^c$ to denote the complimentary set of $\mathcal{I}(x)$. Our results rely on the following assumptions:

**Assumption 2.** *We make the following assumptions on the function $f$ and $h$*

*2.1 Given the initial state $x(0)$ at $t = 0$, the optimization problem in* (11)

$$\lambda = \arg\min_{\lambda \ge 0}\left(\frac{1}{2}\lambda^\top J_h(x)T(x)J_h(x)^\top \lambda \right.$$
$$\left. + \lambda^\top \left(J_h(x)T(x)\nabla f(x) - Kh(x)\right)\right)$$

*admits bounded a solution $\|\lambda\|_\infty \le L$ for all $x \in \mathcal{E}$, where $\mathcal{E}$ is defined by $\mathcal{E} := \{x | 0 < h_i(x) \le h_i(x(0)), \ \forall i \in \mathcal{I}(x(0))\}$.*

*2.2 There exists a constant $M$ such that $\|\nabla f(x)\| < M$, $\|J_h(x)\| < M$ for all $x$.*

*2.3 The function $f(x)$ is lower-bounded, i.e. $f(x) \geq f_{\min}$ for all $x$.*

Although Assumption 2.1 is quite complicated and relatively hard to verify, there are some simplified versions that serve as a sufficient condition of Assumption 2.1. For example, if we start with a feasible $x(0)$, then $\mathcal{E} = \emptyset$ and hence Assumption 2.1 is automatically satisfied. Additionally, note that Assumption 1.1 is another sufficient condition of Assumption 2.1 (see Lemma 3 in Appendix F)

We define the KKT-gap of the state variable $x$ and nonnegative control variable $\lambda \geq 0$ as follows:

$$\texttt{KKT-gap}(x,\lambda) := \max\left\{\|\nabla f(x) + J_h(x)^\top \lambda\|, \left|\lambda^\top h(x)\right|, \max_i [h_i(x)]_+\right\},$$

where $[h_i(x)]_+ = \max\{h_i(x), 0\}$.

**Theorem 4.** *Under Assumption 2, for a diagonal matrix $K = \mathrm{diag}\{k_i\}_{i=1}$, where $k_i > 0$, the learning dynamics (11) satisfies the following properties*

1. *$\frac{dh_i(x(t))}{dt} \leq -k_i h_i(x(t))$, for $i = 1, 2, \ldots, m$, and hence the dynamic will asymptotically converge to the feasible set.*

2. *Define $\ell(x) := f(x(t)) + L\sum_i [h_i(x)]_+$, then $\ell(x(t))$ is non-increasing w.r.t $t$. Here $[h_i(x)]_+ = \max\{h_i(x), 0\}$.*

3. *The following inequality holds*

$$\int_{t=0}^{T} \left(\|\nabla f(x(t)) + J_h(x(t))\lambda(t)\|_{T(x(t))}^2 - \sum_{i \in \mathcal{I}(x)^c} k_i \lambda_i(t) h_i(x(t))\right) dt$$
$$\leq \ell(x(0)) - \ell(x(T))$$

4. *(Asymptotic convergence and convergence rate) The above statements imply that*

$$\inf_{0 \leq t \leq T} \texttt{KKT-gap}(x(t), \lambda(t))$$

$$\leq \max\left\{\sqrt{\frac{2}{\lambda_{\min} T}\left(f(x(0)) - f_{\min} + L\sum_{i \in \mathcal{I}(x(0))} h_i(x(0))\right)}, \right.$$
$$\left. \frac{1}{\min_i k_i} \frac{2}{T}\left(f(x(0)) - f_{\min} + (L+1)\sum_{i \in \mathcal{I}(x(0))} h_i(x(0))\right)\right\}$$
$$\sim O\left(\frac{1}{\sqrt{T}}\right)$$

*Further we have that* $\texttt{KKT-gap}$ *asymptotically converges to zero, i.e.*

$$\lim_{t \to +\infty} \texttt{KKT-gap}(x(t), \lambda(t)) = 0$$

Statement 4 in Theorem 4 implies that the algorithm can find an $\epsilon$-firs-order KKT-point within time $\frac{1}{\epsilon^2}$. Similar to Theorem 1, the key step of the proof is to construct the Lyapunov function in Statement 2 (detailed proof deferred to Appendix D).

## 5. Momentum Acceleration for Constrained Optimization

In Remark 1, we introduced the FL-proximal and FL-Newton algorithms. Generally, FL-Newton achieves faster convergence than FL-proximal due to its use of second-order information. However, in scenarios where Hessian information is unavailable, FL-proximal must be used instead, raising the question of whether its convergence can be accelerated. Given that momentum acceleration has been shown to improve convergence rates in unconstrained optimization, a natural question arises: can a momentum-accelerated version of the FL-proximal algorithm, along with its corresponding discrete-time SQP formulation, achieve faster convergence? This section aims to address this question as part of Contribution 4.

Momentum acceleration is a technique commonly used in optimization to enhance convergence rates (cf. (Polyak, 1964; Nesterov, 1983; d'Aspremont et al., 2021), see Appendix A for more detailed introduction about momentum acceleration). For unconstrained optimization, the discrete-time momentum acceleration for gradient descent generally takes the form of

$$w_t = x_t + \beta(x_t - x_{t-1})$$
$$x_{t+1} = w_t - \eta \nabla f(w_t) \tag{14}$$

Its corresponding continuous-time analogue can be written as a second-order ODE (Polyak, 1964; Su et al., 2016):

$$\dot{x} = z$$
$$\dot{z} = -\alpha z - \nabla f(x) \tag{15}$$

Inspired by the form of (14) and (15), for equality constrained optimization, we propose the following heuristic momentum-accelerated discrete time SQP scheme

$$w_t = x_t + \beta(x_t - x_{t-1})$$
$$\lambda_t = -\left(J_h(w_t) J_h(w_t)^\top\right)^{-1}\left(J_h(w_t)\nabla f(w_t) - \frac{1}{\eta}h(w_t)\right)$$
$$x_{t+1} = w_t + \eta \nabla f(w_t) + J_h(w_t)^\top \lambda_t \tag{16}$$

and continuous time FL scheme, which we name as **FL-momentum**:

---
**FL-momentum for Equality-Constrained Optimization**

$$\dot{x} = z$$
$$\dot{z} = -\alpha z - \left(\nabla f(x) + J_h(x)^\top \lambda\right) \tag{17}$$
$$\lambda = -(J_h(x) J_h(x)^\top)^{-1}(J_h(x)\nabla f(x) - Kh(x))$$
---

Note that the only difference in (16) is that we add a momentum step $w_t = x_t + \beta(x_t - x_{t-1})$. Similarly we can propose the FL-momentum scheme for inequality constraint case as follows:

---

$$\boxed{\begin{array}{l}\textbf{FL-momentum for Inquality-Constrained Optimization}\\[4pt]\dot{x} = z \\[4pt]\dot{z} = -\alpha z - \left(\nabla f(x) + J_h(x)^\top \lambda\right) \qquad\qquad (18)\\[4pt]\lambda = \underset{\lambda \geq 0}{\arg\min}\,\frac{1}{2}\lambda^\top J_h(x)J_h(x)^\top\lambda + \lambda^\top\left(J_h(x)\nabla f(x) - Kh(x)\right)\end{array}}$$

The numerical simulation in Section 6 (Figure 2 and 3) demonstrates the comparison between the standard and momentum accelerated methods, which suggests that momentum methods indeed accelerate the convergence rate.

We would also like to note that as far as we know, the acceleration of SQP methods are generally achieved via Newton or quasi-Newton methods, there's little work on exploring acceleration via momentum approaches, which makes our proposed momentum algorithm a novel contribution.

### 5.1. Analysis

In this section, we provide some convergence guarantees for the proposed algorithm. In particular, we primarily focus on the convergence analysis for the continuous-time algorithm for equality constrained optimization (17). It remains future work to establish the convergence for the discrete-time algorithm (16) or the inequality-constrained algorithm (18).

We first define the following notation

$$\overline{\lambda}(x) := -\left(J_h(x)J_h(x)^\top\right)^{-1}\left(J_h(x)\nabla f(x)\right) \qquad (19)$$

Apart from Assumption 1, we also make the following assumptions on $f$ and $h$.

**Assumption 3.** *Both $f(x)$, $h(x)$ are three-time differentiable and the derivatives are bounded, thus, we know that there exists some constant $L_f, L_1, L_2$ such that*

$$\|\nabla^2 f(x)\| \leq L_f, \quad \left\|\frac{\partial \overline{\lambda}(x)}{\partial x}\right\| \leq L_2,$$

$$\left\|\frac{\partial\left(J_h(x)^\top\lambda(x) + \left(\frac{\partial \overline{\lambda}(x)}{\partial x}^\top h(x)\right)\right)}{\partial x}\right\| \leq L_2$$

**Assumption 4.** *We also assume that that there exists a constant $\bar{H}$ such that*

$$\|\bar{H}(x)\| \leq \bar{H}, \quad \forall x,$$

*where $\bar{H}(x) := [h(x)^\top \nabla^2 h_i(x)]_{i=1}^n$.*

We are now ready to state our main result

**Theorem 5.** *Assume that Assumption 1, 3 and 4 hold. Let two positive constant $a_1, a_2$ be such that*

$$a_2 \geq \left(4\frac{\lambda_{\max}(K)}{\lambda_{\min}(K)}L_2 D + \frac{L_1^2}{\lambda_{\min}(K)}\right) \times a_1 \geq 0.$$

*We define the following Lyapunov function:*

$$\ell(x,z) = a_1\alpha f(x) + \frac{a_2\alpha}{2}\|h(x)\|^2 + a_1\alpha\overline{\lambda}(x)^\top h(x) + \|z\|^2$$

$$+ \left(a_1\nabla f(x) + a_2 J_h(x)^\top h(x) + a_1 J_h(x)^\top\overline{\lambda}(x) + a_1\frac{\partial\overline{\lambda}(x)}{\partial x}^\top h(x)\right)^\top z$$

*then for*

$$\alpha \geq \left(a_1(L_f + L_2) + a_2(M^2 + \bar{H}) + \frac{1}{a_1} + \frac{2(\lambda_{\max}(K)D^2)}{a_2}\right) + 1$$

*we have that*

1. *$\ell(x(t), z(t))$ is non-increasing with respect to t.*
2. *the following inequality holds*

$$\int_{t=0}^{T}\frac{a_2\lambda_{\min}(K)}{8}\|h(x(t))\|^2 + \frac{a_1}{4}\|\nabla f(x(t)) + J_h(x(t))^\top\overline{\lambda}(x(t))\|^2 dt$$

$$\leq \ell(x(0), z(0)) - \min_{x,z}\ell(x,z)$$

3. *We can bound the KKT-gap by*

$$\inf_{0 \leq t \leq T} \texttt{KKT-gap}(x(t), \overline{\lambda}(x(t)))$$

$$\leq \sqrt{\frac{\ell(x(0), z(0)) - \ell_{\min}}{\min\left\{\frac{a_2\lambda_{\min}(K)}{8}, \frac{a_1}{4}\right\}T}} \sim O(\frac{1}{\sqrt{T}})$$

*and*

$$\lim_{t\to+\infty} \texttt{KKT-gap}(x(t), \overline{\lambda}(x(t))) = 0,$$
$$\lim_{t\to+\infty} \texttt{KKT-gap}(x(t), \lambda(t)) = 0.$$

The detailed proof is provided in Appendix E. One limitation of Theorem 5 is that it establishes convergence but not acceleration over FL-proximal. However, when the constraint function $h(x)$ is affine, the algorithm is equivalent to the momentum-accelerated projected gradient method (see Appendix E.1), offering insight into its potential for accelerating optimization.

## 6. Numerical Simulation

### 6.1. Heterogeneous Logistic Regression

In this section, we consider a logistic regression problem involving heterogeneous clients (Shen et al., 2022; Hounie et al., 2024). Many scenarios, such as federated learning and fair machine learning, require training a common model in a distributed manner by utilizing data samples from diverse clients or distributions. In practice, heterogeneity in local data distributions often results in uneven model performance across clients (Li et al., 2020; Wang et al., 2020). Since this outcome may be undesirable, a reasonable objective in such settings is to add constraints to ensure that the model's loss is comparable across all clients.

We formulate the above problem as a constrained optimization problem as follows: consider solving the logistic regression for $C$ clients. For each client $c \in \{1, 2, \ldots, C\}$,

it is associated with its own dataset $D_c = \{(x_i, y_i)\}_{i=1}^{|D_c|}$, where the label is $y_i \in \{-1, 1\}$ and data feature is $x_i \in R^d$. For each client $c$, its own logistic regression loss $R_c(\theta)$ is defined as:

$$R_c(\theta) := \frac{1}{|D_c|} \sum_{i \in D_c} \log(1 + \exp(-y_i \cdot \theta^\top x_i)),$$

where $\theta$ is the parameter of the regression model. We further define the averaged regression loss $\bar{R}(\theta)$ as

$$\bar{R}(\theta) := \frac{1}{C} \sum_{c=1}^{C} f_c(\theta)$$

As suggested in (Shen et al., 2022; Hounie et al., 2024), heterogeneity challenges can be addressed by introducing a proximity constraint that links the performance of each individual client, $R_c$, to the average loss across all clients, $\bar{R}$. This approach naturally formulates a constrained learning problem:

$$\min_\theta \bar{R}(\theta), \quad s.t. \ R_c(\theta) - \bar{R}(\theta) - \epsilon \leq 0, \ \forall c \in \{1, 2, \ldots, C\} \tag{20}$$

where $\epsilon > 0$ is a small, fixed positive scalar.

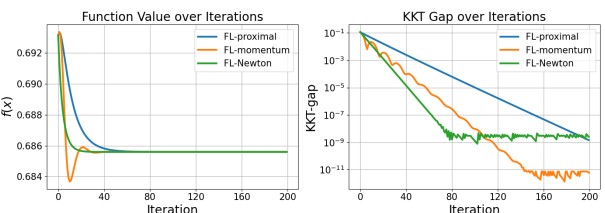

Figure 2: Result for Heterogeneous Logistic Regression

We solve the constrained optimization problem (20) by running the FL-proximal, FL-Newton, and FL-momentum algorithm . Here we set the number of clients to $C = 5$ and $|D_c| = 200$, the data $y_i$ is randomly generated from a Bernoulli distribution and $x_i$ is generated from a Gaussian distribution whose mean differs among different agents. The results of the numerical simulation are presented in Figure 2. Notably, all algorithms successfully converge to a first order KKT point, with FL-Newton exhibiting the fastest convergence, followed by FL-momentum, which outperforms FL-proximal in terms of convergence speed.

### 6.2. Optimal Power Flow

The Alternating Current Optimal Power Flow (AC OPF) problem is a fundamental optimization task in power systems. Its goal is to determine the most efficient operating conditions while satisfying system constraints. This involves optimizing the generation and distribution of electrical power to minimize costs, losses, or other objectives while ensuring that physical laws (such as power flow equations) and operational limits are respected, thus it can be summarized as the following constrained optimization problem:

$$\min_x f(x), \ s.t. \ h_{\text{eq}}(x) = 0, \ h_{\text{ineq}}(x) \leq 0, \tag{21}$$

where the objective function $f(x)$ represents the power generation cost and the equality constraints $h_{\text{eq}}(x)$ generally represents the physical law of the power system, i.e., the power flow equations and $h_{\text{ineq}}$ includes operational limits in terms of voltage, power generation, transmission capacities etc. The optimization variable $x$ generally consists of voltage angles and magnitudes at each bus, and the real and reactive power injections at each generator (see (Low, 2014) for a detailed introduction on AC OPF).

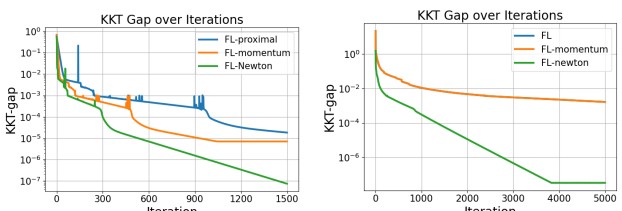

Figure 3: Result for AC OPF on
IEEE-39 bus (left) and IEEE-118 bus (right) bus system

We solve the AC OPF problem (21) by running the FL-proximal algorithm, FL-Newton algorithm, and FL-momentum algorithm. Figure 3 presents the numerical results for solving AC OPF on the IEEE-39 and IEEE-118 bus systems, respectively. In both cases, FL-Newton demonstrates the fastest convergence, which is expected given that it leverages second-order information (i.e., the Hessian). Comparing FL-proximal and FL-momentum, both of which rely solely on first-order information, Figure 3 indicates that FL-momentum accelerates the learning process and achieves faster convergence than FL-proximal for the IEEE-39 bus system. However, in the IEEE-118 bus system, FL-proximal and FL-momentum exhibit similar convergence speeds, with their learning curves nearly overlapping. We hypothesize that this problem is ill-conditioned, limiting the effectiveness of momentum in accelerating the algorithm.

## 7. Conclusion

In this paper, we study the theoretical foundations for solving constrained optimization problems from a control perspective via feedback linearization (FL). We established global convergence rates for equality-constrained optimization, highlighted the relationship between FL and Sequential Quadratic Programming (SQP), and extended FL methods to handle inequality constraints. Furthermore, we introduced a momentum-accelerated FL algorithm, which empirically demonstrated faster convergence and provided rigorous convergence guarantees for its continuous-time dynamics. Future directions include exploring the potential extension to zeroth-order optimization settings and relaxing assumptions in the theoretical analysis.

## Impact Statement

This paper presents work whose goal is to advance the field of Machine Learning, in particular in the domain of control and optimization. There are many potential societal consequences of our work, none of which we feel must be specifically highlighted here.

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
