# OpenReview forum: "Constrained Optimization From a Control Perspective via Feedback Linearization"
_ICML.cc/2025/Conference — Submitted to ICML 2025_

### Official Review · Reviewer_xtqW · 2025-03-10

**Overall Recommendation:** 2

**Summary:**

The paper studies feedback linearization to solve nonconvex optimization problems in which both the objective function and the constraints are nonconvex.

## update after rebuttal
No additional updates

**Claims And Evidence:**

The paper lacks in rigorous definition of key mathematical entities and assumptions necessary for a well-posed optimization framework. For example, the functions defining the optimization problem are not properly specified -- are they continuously differentiable? once? twice? Some assumptions are scattered throughout the paper.

The structure of the paper does not follow the standard flow of papers in these fields: stating the model and its assumption and only then proceeding with the paper, for example, Assumption 1 which facilitates the approach is only stated in the second column of page 3.

The results of the paper rely on restrictive assumptions that are uncommon in practice, and therefore require further justification to be convincing.

**Essential References Not Discussed:**

no

**Experimental Designs Or Analyses:**

I did not see whether the models used in the paper's experiments satisfy the assumptions made in the analysis. If they do not, then these experiments lack meaningful justification -- essentially reducing to running an arbitrary method on an arbitrary problem. When the focus is theoretical, numerical experiments should serve to supplement and complement the theoretical findings, rather than being disconnected from the established assumptions.

**Methods And Evaluation Criteria:**

The paper defines a KKT gap evaluation criteria which is common in this type of challenging models.
However the results and the criteria itself are given in an elaborate manner that is difficult to assess.

**Other Comments Or Suggestions:**

- In line ~89 T(x) is defined as a PD matrix but then you assume boundedness of H(x) which was not defined
- Line 129: don't -> do not
- The paper claims "global convergence" but what is global convergence? sequence? function value? KKT optimality measure? To the best of my knowledge global convergence usually refers to the sequence, and is very hard to attain in nonconvex optimization without restrictive assumptions or sharpness assumption such as the KL property.
- Line 246-247 second column "THe"
- Line 269 second column "bounded a solution"
- Line 325 "firs"

**Other Strengths And Weaknesses:**

Strengths:
- A new perspective on optimizing with nonlinear constraints
- Connection to the well-known SQP method

Weaknesses:
- Confusing structure that makes it hard to follow the framework and approach, consequently the contribution
- Relies on the previous work by Cerone et al. (2024)
- Restrictive assumptions that are not sufficiently justified: LICQ does not usually hold in optimization problems, assuming that the Jacobian has full row rank, and other assumptions
- Missing motivation in the form of explicit model examples in which the assumptions hold
- Advantage over the SQP is not clear

**Questions For Authors:**

none

**Relation To Broader Scientific Literature:**

To the best of my understanding, this paper extends the work of Cerone et al. (2024) and provides an alternative perspective on the SQP method.
However, I am not fully convinced of the added value of its contributions. That said, as I am not sufficiently familiar with the control approach, particularly feedback linearization, I cannot offer a definitive assessment.

**Theoretical Claims:**

I read some of the proofs, but did not fully assessed their correctness.

---

> ### Author Rebuttal · Authors · 2025-04-01
>
> We sincerely thank the reviewer for the comments and suggestions. We have **revised our paper** accordingly (see https://anonfile.io/f/wvtpAhSf). Below, we briefly summarize the reviewer's concerns and then address them one by one:
>
> 1. **Lack of Clarity in Assumptions and Paper Structure:** *The reviewer noted that key assumptions are not clearly or consistently stated, and that convergence criteria (KKT gap) is given in an elaborative manner.*
>
> Response: We thank the reviewer for the feedback. Our intention was to present a streamlined development by stating assumptions only when they are required for specific theoretical results, rather than listing all assumptions upfront. This reflects the fact that the algorithm itself applies broadly to differentiable objective and constraint functions, and that many of the stronger assumptions (e.g., LICQ, smoothness) are only invoked for particular convergence guarantees. That said, we recognize that this design choice may have caused confusion. To improve clarity, we will add a brief clarification at the beginning of the theoretical section stating that all functions are assumed to be differentiable, and that additional assumptions will be introduced where needed to support the analysis.
>
> To further address the reviewer’s confusion in the convergence criteria, we have moved the definition of KKT-gap earlier into the problem formulation part rather than right before stating the theorem.
>
> 2. **Restrictive Assumptions:** *The reviewer found the assumptions underlying the theoretical analysis—such as LICQ and full row-rank Jacobians—too strong or uncommon in practice, and requested better justification and examples where they hold.*
>
> Response: We thank the reviewer for raising this concern. We emphasize that the assumptions invoked in our analysis—such as LICQ (Assumption 1.1)—are standard in the literature on constrained optimization and SQP (see Appendix A paragraph labeed “SQP”). Further, for the inequality-constrained setting, our main convergence analysis relies on Assumption 2.1, which is strictly weaker than LICQ; we clarify this relationship in the discussion following Assumption 2.1 and formally prove it in Appendix F. We also acknowledge that these assumptions may not hold in all practical scenarios; to this end, we include a discussion in Appendix A to highlight their limitations and the potential for extensions beyond the current setting. We would also like to refer the reviewer to our response to Reviewer 3 (Concern 3), who raised a related point.
>
> 3. **Experimental Setup and Justification:** *The reviewer questioned whether the experimental problems satisfy the theoretical assumptions, and found that the connection between the experiments and the theory was not sufficiently clear.*
>
> Response: We thank the reviewer for this helpful observation. We clarify that the models used in our experiments (including Heterogenous Logistic Regression and ACOPF) are constructed to satisfy the standing assumptions in our analysis. In the revised paper, we explicitly state which assumptions are met in each experimental setting.
>
> At the same time, we would like to emphasize that it is often valuable for numerical experiments to test methods beyond the strict confines of theoretical assumptions, as strong empirical performance can indicate the method’s robustness in practice. And indeed, one of the strengths of our approach is that it can be applied without explicitly checking strong conditions.
>
> 4. **Relation to SQP and Novelty:** *The reviewer felt that the contribution appears closely tied to prior work such as Cerone et al. (2024) and was not fully convinced of the added value or novelty of the proposed approach over established SQP methods.*
>
> Response: We thank the reviewer for raising this point. We would like to emphasize several aspects of our contribution that we believe are novel and meaningful.
>
> First, in contrast to prior works such as Cerone et al. that provide only local stability guarantees in the nonconvex setting, our analysis establishes non-asymptotic global convergence rates. This represents a nontrivial advancement, particularly given the complexity of constrained, nonconvex dynamics.
>
> Secondly, compared with the SQP algorithm, we proposed a novel momentum algorithm, which is different from classical SQP formulations and is able to accelerate convergence in using only first-order information. Additionally, compared with the SQP algorithm, the control-theoretic lens we adopt—particularly through feedback linearization—offers a broader design framework. While our current work targets a specific dynamic, this viewpoint allows the incorporation of more general stabilizing controllers for constraint satisfaction, potentially leading to new classes of algorithms with desirable robustness or geometric properties. We would also like to point the reviewer to our response to Reviewer 3 (Concern 5), who raised a related point.

---

> > ### Comment · Reviewer_xtqW · 2025-04-03
> >
> > Thank you for the clarifications and updates.
> > I was able to briefly survey the new version you provided and it indeed looks better in structure and writing, IMO.
> > However, I cannot provide another review to a revised version of the paper, and so cannot significantly reassess my recommendation based on the the revised version.
> >
> > Considering the rebuttal and the other reviewers comments, and since I am not sufficiently familiar with the field and literature, I am keeping my recommendation as is.

---

### Official Review · Reviewer_QUgP · 2025-03-10

**Overall Recommendation:** 2

**Summary:**

The manuscript proposes a new perspective on analyzing first-order algorithms in constrained optimization that is rooted in the control-theoretic notion of feedback linearization. The manuscript is overall well written and the main innovation is presented well. There are also interesting numerical examples that span distributed logistic regression and the computation of power flow in well-known standard benchmarks (IEEE standard). The topics are of interest to the ML community.

In my view the strength of the work lies in introducing the concept of feedback linearization to the optimization/machine learning community. I do think that the control-theoretic notion of feedback linearization - or more generally, finding nonlinear coordinate transformations resulting in linear dynamics (e.g. Frobenius theorem / differential topology) - could be very valuable for understanding optimization/training dynamics, and is certainly an underrepresented branch of mathematics in the optimization/computer-science/learning community. However, I am not convinced that the article achieves this goal and I also have doubts whether the setting analyzed in the article is appealing/promising. On the one hand, there is a body of already existing work analyzing similar dyanmics and on the other hand, the work considers the feedback linearization of y=h(x(t)), which is only a subset of the relevant dynamics.

Specific comments:
- There are numerous control-theoretic interpretations for the dynamics in (3), where the dual variable is considered to be a control input, including [1], [2]. The authors could possible further strengthen the narrative of why the "feedback-linearization" prespective is useful/fruitful/novel and adds value to the analysis of gradient-based algorithms.

- From an algorithmic perspective the question arises whether the authors can derive new/easier/more genearl results for the resulting *discrete-time* systems. From a mathematical/technical perspective, deriving precise rates for (discrete-time) algorithms is typically much harder than the continuous their continuous-time approximations. Hence, from a purely technical point of view, the manuscript has limited added value to the optimization/learning community. Is there a way the feedback-linearization perspective could help/facilitate the discrete-time analysis?

- The fact that the authors assume a unique solution for the optimization over dual variables in (11) seems strong from an optimization perspective. Can the authors connect the assumption to constraint qualification? Please note that LICQ is restrictive.

- Similarly, it would be helpful if the authors could discuss Assumption 2.1 (bounded multiplier) in the context of constraint qualification. Why is the relatively strong assumption that lambda is unique not listed in Assumption 2?

- The authors might want to justify why continuous-time rates are meaningful, since in principle, reparametrization of time can be used to speed up convergence.

- The authors might want to justify why the analysis is focused on deriving slow 1/sqrt(T) rates to stationary points in a nonconvex setting. Often, deriving precise accelerated/non-accelerated rates, e.g. 1/t, 1/t^2, etc. for the convex setting is much more challenging. This would also provide further insights on whether acceleration is actually achieved when adding momentum.

- As a result of the feedback linearization of y=h(x(t)) the work suggests to implement a linear decrease in constraint violation of the type dh(x(t))/dt=- k h(x(t)), where k is constant. Please note that there has been quite a few works that suggested similar approaches, which seem relevant and should possibly be cited, in particular [3], [4], [5] (see also references therein). Most works in the ML community prove discrete-time rates even in the momentum-based setting [5] or consider optimization over manifolds [4] where retractions are challenging to compute.

- The claim in the 4th contribution "To the best of our knowledge, both the proposed algorithm and its analysis are novel contributions to the field" needs to be revised in light of the state-of-the art in the literature. Note that [5] provides precise accelerated rates in *discrete-time*.

- The paper could benefit from some polishing and comes across as sloppy. There are colloquial terms such as "don't", the punctuation is not consistent with the standards of mathematical writing, and the typesetting of references is funky (just to name a few typographical things).


[1] J. Wang and N. Elia, "A control perspective for centralized and
distributed convex optimization," CDC, 2011
[2] A. Allibhoy and N. Cortes, "Control-Barrier-Function-Based Design of Gradient Flows for Constrained Nonlinear Programming," IEEE TAC, 2024
[3] M. Muehlebach and M. I. Jordan, "On Constraints in First-Order Optimization: A View from Non-Smooth Dynamical Systems," JMLR, 2022
[4] P. Ablin and G. Peyre, "Fast and accurate optimization on the orthogonal manifold without retraction," PMLR, 2022
[5] M. Muehlebach and M. I. Jordan, "Accelerated First-Order Optimization under Nonlinear Constraints," arXiv:2302.00316, 2023

**Claims And Evidence:**

see above.

**Essential References Not Discussed:**

see above.

**Experimental Designs Or Analyses:**

see above.

**Methods And Evaluation Criteria:**

see above.

**Other Comments Or Suggestions:**

see above.

**Other Strengths And Weaknesses:**

see above.

**Questions For Authors:**

see above.

**Relation To Broader Scientific Literature:**

see above.

**Theoretical Claims:**

see above.

---

> ### Author Rebuttal · Authors · 2025-04-01
>
> We thank the reviewer for the constructive feedback. We have incorporated the suggestions into the **revised manuscript** (see https://anonfile.io/f/wvtpAhSf). We summarize and address the reviewer’s concerns as follows:
>
> 1. **Feedback linearization perspective:** *How does the feedback linearization viewpoint compare to existing methods, such as those in [1] and [2] cited by the reviewer?*
>
> Response: We thank the reviewer for highlighting the question. For this part, we would like to refer the reviewer to the Response to Reviewer 1 (Concern 1) who raised the same concern, as well as the Response to Concern 5 below.
>
> The reviewer also notes that our analysis focuses on the feedback linearization of y=h(x(t)), which only captures a subset of the dynamics. This is precisely why establishing global convergence guarantees is crucial: while feedback linearization ensures constraint satisfaction, it does not in itself imply stability or convergence; thus, developing convergence guarantees serves as one of the key contributions of our paper.
>
> 2. **Continuous vs. discrete-time analysis**: *The reviewer questions the value of continuous-time rates and whether the feedback linearization perspective can facilitate discrete-time analysis.*
>
> Response: We agree that continuous-time analysis has limitations as the reviewer suggested. However, even in continuous time, analyzing the dynamics in nonconvex settings is challenging and meaningful in its own right. Our continuous-time framework serves two purposes: (i) it offers theoretical insight into stability and convergence, and (ii) the discretization of the continuous-time dynamics directly informs discrete-time design. In this paper, we show that forward Euler discretization of the feedback-linearized dynamics yields an update closely related to SQP algorithm in discrete time. Moving forward, we believe it would be highly valuable to further explore the discrete time convergence rate and how different discretization schemes (e.g., higher-order methods) affect convergence both in theory and in practice.
>
> 3. **Assumption justification:** *Clarification is requested regarding the necessity of the assumptions, particularly the uniqueness of the dual solution and Assumption 2.1, and its connection to constraint qualification.*
>
> Response: We thank the reviewer for raising this point. Our analysis relies on Assumption 2.1, which requires the existence of a bounded (not necessarily unique) dual variable; the uniqueness mentioned after Equation (11) is used solely for the ease of demonstration and does not affect the proofs. We have revised the manuscript to clarify our assumptions.
> Regarding the connection to constraint qualifications: we realize that Assumption 2.1 is closely connected to the MFCQ assumption in [3,5], which is weaker than LICQ, we have revised our discussion of the assumption (highlighted in blue).
>
> 4. **Acceleration and rate focus:** *Why does the paper focus on the slower $1/\sqrt{T}$ convergence rate in a nonconvex setting, rather than sharper rates (e.g., $1/T$, $1/T^2$) in convex settings? How does this relate to acceleration?*
>
> Response: We would like to refer the reviewer to the Response to Reviewer 1 (Concern 2) who raised the same concern.
>
> 5. **Novelty of the accelerated methods:** *The reviewer suggests revisiting the claim of novelty in light of recent related works, including [2,3,4,5].*
>
> Response: We sincerely thank the reviewer for pointing us to relevant works! Note that [4] primarily focuses on the setting where the constraint is an orthogonal manifold, so the setting is different from ours. While the algorithms in [2,3,5] are admittedly similar in spirit to ours, we would like to highlight a few key differences that underscore our contributions.
>
> - In the **nonconvex setting**, [2,3,5] provide asymptotic convergence guarantees, whereas our work establishes **non-asymptotic convergence rates**. In particular, our analysis is not a trivial extension. In particular, we construct a **Lyapunov function** tailored to the dynamics, which is not considered in  [2,3].
> - We establish a connection of the method to the classical SQP algorithm, a perspective not explored in [2,3,5].
> - While [5] and our momentum method share some similarities, both the continuous-time dynamics and the discretization approach differ, resulting in distinct algorithms. In the revised paper, we have updated our claims, acknowledged [5], and added a detailed comparison highlighting both the differences and similarities.
>
> Furthermore, we believe feedback linearization provides a more **general framework** beyond specifying a linear target of the form \dot{y} = -Ky​. In principle, this approach allows us to leverage **arbitrary stable controllers, such as PI controllers** for constraint enforcement, potentially offering new algorithmic behaviors or robustness benefits. We have added corresponding additional results in **Remark 2 and Appendix B.2**.

---

### Official Review · Reviewer_h2iB · 2025-03-12

**Overall Recommendation:** 4

**Summary:**

The paper develops the view of optimization methods as (optimal) feedback control of ODEs, which is some ways is the original view in the Soviet literature, but has been rediscovered only in the past decade in the Western machine learning, with the work of Andrew Packard, Bin Hu and others. Among others, it allows for inequalities to be considered in the optimization problem.

**Claims And Evidence:**

The main idea is nicely argued, but some parts of the manuscript seem to have been written in a rush.

Issues:
- line 93 on the right: H(x) never reappears in the paper and it is in the same sentence as T(x), shouldn’t it be T(x) then?
- line 97 on the right: is there a lambda missing in the equation \nabla f(x^*) + J_h^T(x^*) = 0?
- line 325 on the right: what does “Note that the only difference in (16) is that we add a momentum step w_t = x_t +\beta(x_t −x_{t−1} )” - - - mean? The only difference as opposed to what?
- line 367 on the left: there should be L_1 instead of L_2
- line 356-357 on the right: inconsistency in notation of \lambda and \bar\lambda with respect to the variable (t or x(t))
- Figure 3: the label of the blue line “FL-proximal” is missing

**Essential References Not Discussed:**

None, afaik.

**Experimental Designs Or Analyses:**

The experiments seem illustrative.

**Methods And Evaluation Criteria:**

Isn’t it weird that the blue line on the right of Figure 3 is missing completely? The authors claim that it “nearly overlaps with the FL-momentum curve”, but it seems improbable that we can’t see it at all.

**Other Comments Or Suggestions:**

A proof-reading would help.

**Other Strengths And Weaknesses:**

Minor issues:
- line 96 on the right: it must satisfy >> must satisfy
- line 141 on the right: "which is equivalent to Jh (x)Jh (x)^T is invertible" >> "which is equivalent to Jh (x)Jh (x)^T being invertible"
- line 219 on the left: “in to Appendix A” >> “to Appendix A”
- line 242 on the left: “theses” >> “these”
- line 246 on the right: “THe”
- line 257 on the right: “complimentary set” wrong spelling, moreover this is redundant as this notation is introduced on page 2
- line 259 on the right: “admits bounded a solution”
- line 325 on the left: “an ϵ-firs-order”
- lines 363-365 on the left: “three-time differentiable” >> “three-times differentiable; “there exists some constant” >> “there exist some constants”
- line 381 on the left: “two positive constant” >> “two positive constants”

**Questions For Authors:**

None.

**Relation To Broader Scientific Literature:**

There is a good overview of the literature.

**Theoretical Claims:**

- Theorem 1 could be stated more neatly. E.g. “Under Assumption 1, for
control gain K that is a diagonal positive definite matrix, i.e., K =
diag{k_i} , where k_i > 0, we have that…” >> “Let Assumption 1 hold
and let the control gain K be a diagonal positive definite matrix,
i.e., K = diag{k_i} , where k_i > 0. Then…”

- line 265 on the right: The optimization problem in (11) “lambda =
argmin…” is stated on this page and then repeated twice on the same
page. Labeling the equations in (11) separately as (11.1) and (11.2)
would solve these redundancies.

- The assumption 1.2 coincides with 2.2 and the assumption 1.3 coincides
with 2.3. Having 1.2 and 1.3 as separate assumptions would solve these
redundancies.

- In general: inconsistent use of colons before equation terms and full
stops after them. E.g., compare line 234-238 on the left and on the
right.

---

> ### Author Rebuttal · Authors · 2025-04-01
>
> We sincerely thank the reviewer for carefully reading our paper and providing constructive feedback, as well as for their positive assessment of our work and its positioning within the broader literature on optimization and control. We are also grateful for the reviewer’s detailed suggestions on typos and improvements to the phrasing of theorems and statements, which have helped us enhance the clarity of the manuscript. With regard to the reviewer’s comments on typos or notational inconsistencies, if we did not provide a specific response, it means we fully agree with the reviewer’s observation and have corrected the issue accordingly in the **revised manuscript** (see https://anonfile.io/f/DEcRCSxS). Below, we address the technical questions raised by the reviewer.
>
> 1. **Elaboration on FL-momentum**: *"line 325 on the right: what does “Note that the only difference in (16) is that we add a momentum step w_t = x_t +\beta(x_t −x_{t−1} )” - - - mean? The only difference as opposed to what?"*
>
> Response: We sincerely apologize for the confusion. In this part, we intended to compare Equation (16) (now Equation (18) in the revised version) with Equation (6) (now Equation (8)), where FL-proximal is used without momentum. We have revised the text for clarity as follows: “Note that compared with FL-proximal (8), the difference in (18) is the addition of a momentum step $w_t = x_t + \beta(x_t - x_{t-1})$.”
>
> We hope this revision helps clarify the point, and we truly appreciate the reviewer’s feedback. We would be happy to further discuss this if there are any remaining questions.
>
> 2. **Question about Figure 3**: *“Isn’t it weird that the blue line on the right of Figure 3 is missing completely? The authors claim that it “nearly overlaps with the FL-momentum curve”, but it seems improbable that we can’t see it at all. ”*
>
> Response: We sincerely thank the reviewer for carefully examining the details of our numerical experiments. In the IEEE 118-bus example, we observed that the algorithm becomes unstable when the momentum hyperparameter \( \beta \) is set too large. The best-performing value found during hyperparameter tuning was \( \beta = 0.02 \), which results in the FL-proximal and FL-momentum curves appearing nearly identical. We believe this is due to the ill-conditioned optimization landscape of the problem, which limits the benefit of momentum. To clarify this, we have plotted a zoomed-in version of the learning curves (see here https://anonfile.io/f/huBzM42f), where subtle differences between the two methods can be observed. We greatly appreciate the reviewer’s observation and have double-checked the plot to ensure its accuracy.

---

### Official Review · Reviewer_vVQJ · 2025-03-15

**Overall Recommendation:** 2

**Summary:**

The paper develops a theoretical foundation for using feedback linearization (FL) from control theory to address constrained optimization problems, proving global convergence rates, extending FL methods to inequality constraints, relating FL to Sequential Quadratic Programming (SQP), and introducing a novel momentum-accelerated FL algorithm with proven convergence guarantees.

**Claims And Evidence:**

In the abstract you mention that
> Furthermore, we introduce a momentum accelerated FL algorithm that achieves faster convergence, and provide a rigorous convergence guarantee.

This seems misleading to me, since I would expect you show a faster convergence rate, which you didn't.

**Essential References Not Discussed:**

Lack discussions on the convergence rates of common first-order methods.

**Experimental Designs Or Analyses:**

I don't see any explicit analysis or comparison of computational complexity, wall-clock time comparisons or memory requirements in their experimental section. Scalability with problem size should also be analyzed.

**Methods And Evaluation Criteria:**

My main concern is the method requires matrix inversion and thus belongs to the second-order genre. However, a $O(1/\sqrt{T})$ is given, which could be achieved by common 1st-order methods such as projected gradient descent, or Augmented Lagrangian Methods, which also handle constraints naturally. Nesterov's Accelerated Gradient Descent can achieve $O(1/T^2)$ convergence rate, but this paper doesn't have a rate for their accelerated variant. And do to the need of matrix inversion, the proposed algorithms are hard to scale up.

**Other Comments Or Suggestions:**

see above.

**Other Strengths And Weaknesses:**

See above.

**Questions For Authors:**

see above.

**Relation To Broader Scientific Literature:**

This paper aims to bridge control theory, dynamical systems, and optimization by establishing theoretical foundations for feedback linearization in constrained optimization.

**Theoretical Claims:**

The proofs seem correct to me. However:

- The convergence rate is slow for a 2nd-order method. See also my comment in Methods And Evaluation Criteria.

- In Theorems 1 and 4, the authors establish convergence rates of $O(1/\sqrt{T})$ for their algorithms. However, they don't explicitly characterize how these rates depend on other important problem parameters like the dimensionality $n$, the number of constraints $m$, condition numbers, or properties of the objective function and constraints (the constants appear in the assumptions).

- Assumption 3 (the three-times differentiability for momentum analysis) seems less standard and a bit strong.

---

> ### Author Rebuttal · Authors · 2025-04-01
>
> We sincerely thank the reviewer for the comments and suggestions! We have **revised our paper** (see https://anonfile.io/f/DEcRCSxS) accordingly. We briefly summarize the reviewer’s main concerns as follows and address them one by one:
>
> 1. **Missing Discussion on First-Order Methods:** *The reviewer suggested that the paper should include a more explicit comparison with the convergence rates of standard first-order methods, such as primal-dual gradient descent (PDGD), projected gradient descent (PGD), and augmented Lagrangian methods (ALM).*
>
> Response: We thank the reviewer for raising this point. We would like to point the reviewer to **Appendix A** for a detailed comparison with existing first order methods, which we briefly summarize as follows:
>
> PDGD has been extensively studied (e.g., Kose, 1956; Wang and Elia, 2011; Qu and Li, 2018), but is primarily restricted to convex settings. Several works suggest that PDGD can fail to converge or behave poorly in nonconvex problems. In contrast, our feedback linearization framework remains applicable in nonconvex settings, as supported by our theory and experiments.
> PGD also becomes unsuitable in nonconvex problems, as projections onto general nonconvex sets are often computationally intractable.
>
> While ALM can handle nonconvex constraints, each iteration involves solving a nonconvex subproblem, which may be costly. Moreover, in settings where the constraint dimension is much smaller than the dimension of 𝑥, SQP-based approaches (such as ours) tend to perform better numerically.
>
> In the revised paper, we have added **corresponding numerical results** to compare these methods in Appendix B.1. We thank the reviewer for helping improve the clarity and reliability of our presentation.
>
> 2. **Claim About Acceleration and Convergence Rate:** *The reviewer found the abstract potentially misleading, as it mentions acceleration without demonstrating an improved convergence rate over the baseline method.*
>
> Response:  We thank the reviewer for this helpful observation and apologize for the ambiguity in the current phrasing. As noted in our response to Concern 1, a key strength of the feedback linearization (FL) framework is its applicability to nonconvex problems. In such settings, even momentum-accelerated methods are only known to achieve a convergence rate of $O(1/\sqrt{T})$, which is the rate we aim to establish in this work.
>
> Although convex optimization is not the focus of the paper, we note that our accelerated algorithm can be connected to **momentum-accelerated projected gradient descent** in the convex setting (see **Appendix E.1**). This connection allows us to leverage existing convergence results for projected accelerated methods, providing theoretical justification for the observed acceleration of our algorithm in the convex regime. We have revised the claim in our paper to clarify the confusion (Remark 5).
>
> 3. **Second-Order Nature and Scalability**: *The reviewer noted that the algorithm requires matrix inversion, placing it closer to second-order methods, and raised concerns about its scalability to high-dimensional problems.*
>
> Response: We thank the reviewer for raising this point. First, we would like to clarify that although our methods (FL-proximal and FL-momentum) involve matrix inversion, they are not second-order methods, as they only require first-order information—specifically, the gradient of $f$ and the Jacobian of $h$; no Hessians or second-order derivatives are used.
>
> Second, regarding scalability: the matrix inversion arises in a subspace defined by the constraints. In many practical settings, e.g. safe RL, the number of constraints is significantly smaller than the dimension of $x$. In such cases, the inversion is computationally inexpensive and can be performed efficiently. We will clarify this in the revision and emphasize that the cost of inversion depends on the number of constraints, not the ambient dimension.
>
> We appreciate the reviewer’s comment on scalability, and we have incorporated the above discussion in the revised manuscript (Remark 1).
>
> 4. **Lack of Computational Complexity Analysis**
>
> Response: We thank the reviewer for this observation and have added Remark 1 to discuss computational complexity in the revised manuscript.
>
> 5. **Convergence Rate Dependencies:** *The reviewer observed that the convergence rates established in Theorems 1 and 4 do not explicitly characterize dependencies on key problem parameters, such as dimensionality, number of constraints, or condition numbers.*
>
> Response: We appreciate the reviewer’s comment on this point. The convergence rates depend not directly on the dimensionality or number of constraints, but on parameters such as M and D (from Assumption 1), and the condition number of the matrix T(x). (See e.g. the detailed convergence rate formula in Statement 4 within Theorem 1 and 4).
>
> Once again, we appreciate the reviewer’s feedback and are open to further discussions.

---

### Decision · Program_Chairs · 2025-05-01

**Decision:**

Reject

**Comment:**

This paper proposes a feedback linearization-based method in a continuous time scale to solve constrained nonconvex optimization problems. The authors also extend it to handle inequality constraints, establish connections with SQP and introduce a momentum-accelerated variant.

Strengths:

The work introduces feedback linearization as a new analytical lens for constrained optimization. The theoretical development connects classical control concepts with optimization theory and provides convergence guarantees, even under nonconvex constraints.

Weakness:

The discrete-time algorithmic implications remain underdeveloped. While the continuous-time analysis is mathematically sound, its practical relevance to optimization is unclear without an accompanying discrete-time analysis. In fact, to the best of Area Chair's knowledge, the state-of-the-art convergence rate for non-convex optimization with non-convex inequality constraints satisfying a LICQ-type assumption is $O(\frac{1}{T^{1/3}})$. This rate has not been improved for a few years and it is an open problem if this can be improved. However, the continuous-time analysis in this paper leads to $O(\frac{1}{T^{1/2}})$. This suggests either the continuous-time convergence rate cannot be preserved when implemented in the discrete-time setting, or there is a way to improve the $O(\frac{1}{T^{1/3}})$ rate in the discrete-time case (which the authors do not analyze).

I think the paper can be stronger if the authors derive the corresponding discrete time convergence rate. If the rate is still for $O(\frac{1}{T^{1/2}})$ for the non-convex inequality constrained case, this paper will improve the state-of-the-art.

Several reviewers noted missing or incomplete comparisons to related work. One review is concerned that the experimental section lacks grounding in the theoretical assumptions.